# Tuberculosis notification in a private tertiary care teaching hospital in South India: a mixed-methods study

Archana Siddaiah,[1] Mohammad Naseer Ahmed,[2] Ajay M V Kumar,[3,4]
George D'Souza,[5] Ewan Wilkinson,[6] Thae Maung Maung,[7] Rashmi Rodrigues[1,8,9]

For numbered affiliations see end of article.

**Correspondence to**
Dr Archana Siddaiah;
archanapink@yahoo.com

## ABSTRACT

**Objectives** India contributes approximately 25% of the 'missing' cases of tuberculosis (TB) globally. Even though ~50% of patients with TB are diagnosed and treated within India's private sector, few are notified to the public healthcare system. India's TB notification policy mandates that all patients with TB are notified through Nikshay (TB notification portal). We undertook this study in a private hospital to assess the proportion notified and factors affecting TB notifications. We explored barriers and probable solutions to TB notification qualitatively from health provider's perspective.

**Study setting** Private, tertiary care, teaching hospital in Bengaluru, South India.

**Methodology** This was a mixed-methods study. Quantitative component comprised a retrospective review of hospital records between 1 January 2015 and 31 December 2017 to determine TB notifications. The qualitative component comprised key informant interviews and focus groups to elicit the barriers and facilitators of TB notification.

**Results** Of 3820 patients diagnosed and treated, 885 (23.2%) were notified. Notifications of sputum smear-positive patients were significantly more likely, while notifications of children were less likely. Qualitative analysis yielded themes reflecting the barriers to TB notification and their solutions. Themes related to barriers were: (1) basic diagnostic procedures and treatment promote notification; (2) misconceptions regarding notification and its process are common among healthcare providers; (3) despite a national notification system other factors have prevented notification of all patients; and (4) establishing hospital systems for notification will go a long way in improving notifications.

**Conclusions** The proportion of patients with TB notified by the hospital was low. A comprehensive approach both by the hospital management and the national TB programme is necessary for improving notification. This includes improving awareness among healthcare providers about the requirement for TB notifications, establishing a single notification portal in hospital, digitally linking hospital records to Nikshay and designating one person to be responsible for notification.

## BACKGROUND

In 2016, approximately 40% of the estimated 10.4 million tuberculosis (TB) cases were

## Strengths and limitations of this study

► A mixed-methods design where the qualitative component explains and complements the findings from the quantitative component.
► Retrospective nature of the quantitative component ensured that the study procedures did not influence the notifications.
► It is likely that both the proportions notified and the number of patients diagnosed or treated are marginal overestimates.
► The findings are limited by the quality of the records maintained.

'missing', that is, were undiagnosed or unreported.[1] India contributes approximately 25% of the 'missing' cases globally.[1 2] Finding these 'missing' cases and treating them successfully is vital to ending TB by 2030, as envisaged by the United Nations Sustainable Development Goals.[3 4]

Healthcare delivery in India involves both the public and private sectors. The Indian private healthcare sector is estimated to cater to approximately two-thirds of the inpatients and three-fourths of the outpatients in the country.[5] The private healthcare sector also accounts for 54% of the healthcare teaching facilities in India.[5] It is therefore not surprising that approximately two-thirds of the 2.2 million patients with TB annually are diagnosed and treated within the private healthcare sector.[6] However, in 2017 only 19% of these patients receive care from, or are notified, that is, reported, to the Revised National Tuberculosis Control Programme (RNTCP),[4 7] India's national health programme for the prevention and control of TB, as compared with 81% from public sector. Though mandatory TB notification introduced in 2012 saw a sharp increase in TB notifications, notification from the private sector continues to be low.[4 7–11] This is despite launching Nikshay, the case-based

web-based national TB notification portal, accessible to all healthcare providers, laboratories and diagnostic facilities, both public and private, nationwide.

Improving the estimates of disease prevalence though is essential for planning, monitoring and evaluation of RNTCP. Yet barriers such as lack of time, poor awareness regarding notification, concern about breaching patient confidentiality, operational complexities in notifying, along with lack of trust in the public sector prevent complete notification.[12–14]

The information on the extent of notification from private tertiary care teaching facilities is limited. This study was designed to determine the proportion of TB cases notified and the factors that affect notification in a private tertiary care teaching healthcare facility in Karnataka State, South India. The study also explored qualitatively the gaps in the existing notification systems so as to enable the identification and development of strategies to improve notification.

## METHODS
### Study design
A mixed-methods study comprising a retrospective review of records to quantitatively assess the proportions of patients with TB notified, and a qualitative component to identify barriers to TB notification was used.

### Study setting
The study was conducted at a private tertiary-level teaching hospital in Bengaluru, Karnataka State, South India. The hospital has 1250 beds and caters to approximately 2000 outpatients daily from diverse backgrounds. A network of laboratory, pathology and radiology services support the clinical departments at the hospital. TB-specific microbiological services available are microscopy, GenXpert MTB/RIF, solid culture, and liquid TB culture and drug susceptibility testing (such as mycobacterial growth indicator tube).

There is a computerised information system for these services and the pharmacy exists at the hospital. The Medical Records Department compiles and maintains inpatient and outpatient hospital records in paper format. Inpatient records are available electronically and outpatient records are available in paper format.

### The Indian RNTCP and its relationship with the study hospital
The RNTCP, a vertical national health programme, strives to provide care and treatment at no cost to all patients with TB in India. The programme adheres to the diagnostic and treatment recommendations of the WHO.[15] The programme delivers its services through a network of designated microscopy centre (DMC, population covered: 0.1 million) and peripheral health institutions (PHI) (primary, secondary and tertiary healthcare facilities including all healthcare academia).[16]

In addition, direct observed treatment (DOT) centres at PHIs are responsible for dispensing treatment, observing treatment doses swallowed (DOT), patient follow-up and patient retention in care. Until 2017, the RNTCP followed an alternate day treatment regimen, with DOT thrice a week in the intensive phase (2 months) and weekly once in the continuation phase (4 months). All public PHIs function as DOT centres and have a TB health visitor (TBHV), responsible for DOT and patient retention. DOT centres at academic institutions, however, have a medical officer in addition to the TBHV. A PHI may also function as DMC.

Even though the RNTCP sets guidelines it does not dictate diagnostic or treatment protocols to the private sector. However, it attempts to deliver public services to the private sector through public-private partnerships (PPP) and expects all private healthcare providers to notify patients with TB irrespective of a PPP through Nikshay.

### Management of TB at the hospital
By virtue of being a private tertiary care teaching hospital the RNTCP has established a DMC and a DOT centre at the hospital through a PPP. The RNTCP staff at the study hospital therefore comprised aLaboratory technician (LT), a Medical Officer (MO) (position currently vacant) and a TBHV.

When diagnosed with TB at any of the clinical departments at the hospital, patients can choose to take anti-tubercular treatment (ATT) either through the DOT centre, at no cost, or through the hospital's pharmacy for a cost. The patient's physician guides the patient's choice of treatment on a case by case basis.

### Notification of patients with TB at the study hospital
Irrespective of the source of treatment, all patients with TB who are diagnosed or treated at the hospital are expected to be referred to the DOT centres for registration with the RNTCP and subsequent notification via the online notification portal Nikshay. In the study hospital notification of patients with TB was the responsibility of the TBHV.

### Study population
#### Quantitative component
Study subjects comprised all patients diagnosed with TB and/or treated for TB from 1 January 2015 to 31 December 2016. For this study, the definition of a patient with TB incorporated the RNTCP definitions and patients identified through pharmacy records. Pharmacy records served as a surrogate, especially for the outpatients diagnosed, in absence of outpatient electronic health records at the hospital. A patient with TB was therefore defined as (1) microbiologically confirmed (RNTCP): a patient with microbiologically confirmed TB using microscopy, bacterial culture and/or GenXpert MTB/RIF, or (2) clinically diagnosed (RNTCP): a patient with histopathological or radiological findings suggestive of TB, irrespective of microbiological confirmation, or (3) a patient who availed

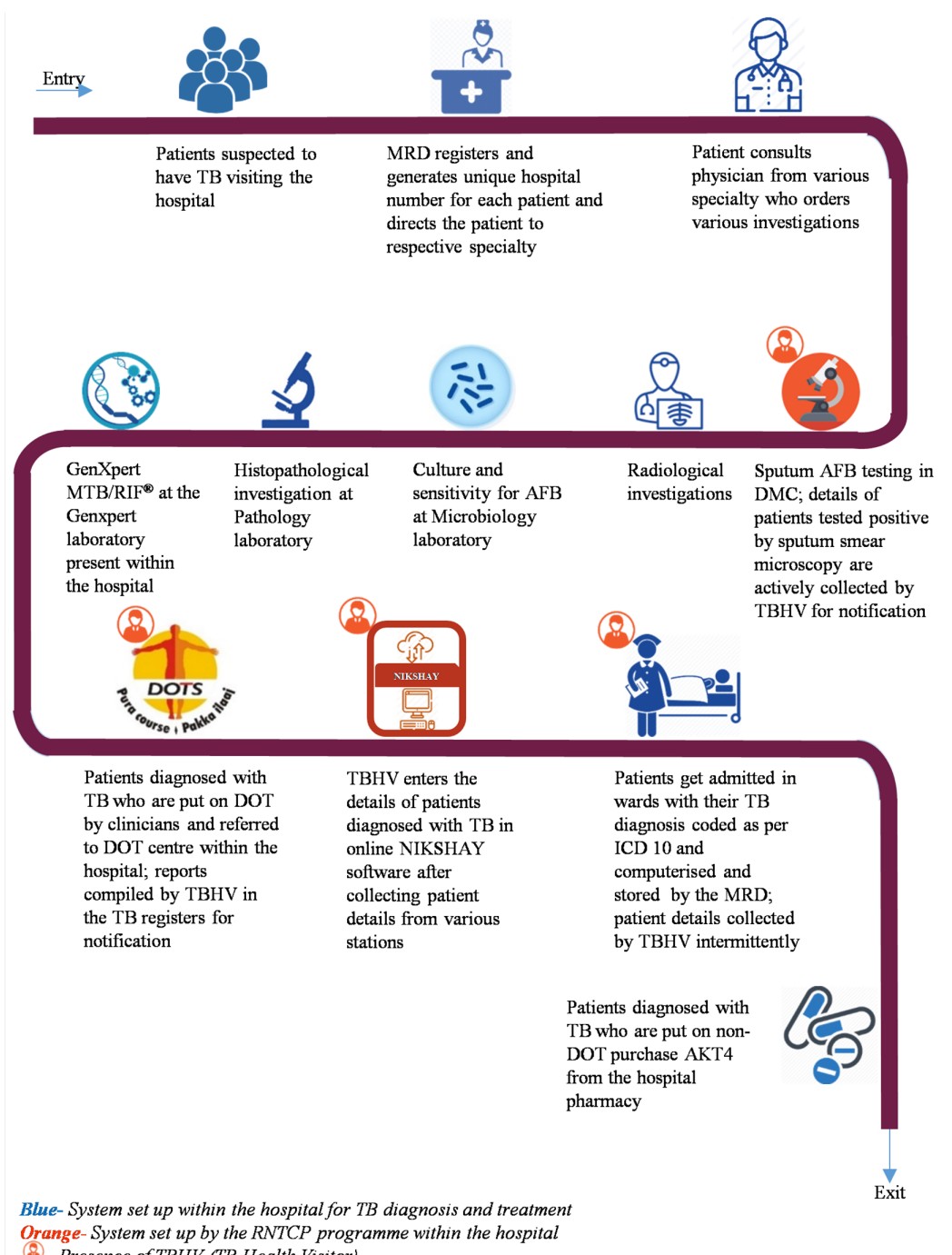

**Blue**- System set up within the hospital for TB diagnosis and treatment
**Orange**- System set up by the RNTCP programme within the hospital
🧑 - Presence of TBHV (TB Health Visitor)

**Figure 1** Flow of patients seeking TB care at a private tertiary care teaching hospital in Bengaluru, India. AFB, acid-fast bacilli; AKT4, anti-TB medication; DOT, direct observed treatment; ICD-10, International Classification of Diseases 10th Revision; MRD, Medical Records Department; RNTCP, Revised National Tuberculosis Control Programme; TB, tuberculosis; TBHV, TB health visitor.

ATT from the hospital's pharmacy identified through the pharmacy information system (PIS).

### Qualitative component
Healthcare providers caring for patients with TB from various departments including clinicians, staff nurse, researchers, LT and TBHV were interviewed in-depth. Participants were chosen purposively to include those

involved at various points within the TB case management cascade which is depicted in figure 1.

### Data sources, variables and data collection procedures
### Quantitative component
Demographic details of patients with TB such as patient's name, date of birth, gender, education, marital status, and residence (urban/rural), and year diagnosed, clinical department visited and source of the record were

extracted from multiple sources. Data were first extracted from the inpatient electronic medical records database using the International Classification of Diseases 10th Revision (ICD-10) coding for TB (codes A15–A19). Subsequently data from the histopathology component of the laboratory information system (LIS) were extracted. For this, search terms such as 'tuberculosis', 'TB' and for possible typographical errors and 'lower and upper case formats' (eg, TB or tb) were used, as these diagnoses did not follow the ICD-10 coding. Data were similarly extracted from the radiology information system (RIS). These comprised reports from CT and MRI. Chest radiographs were not reported in the RIS as physicians review them in the light of clinical evidence for diagnosis. A laboratory or radiology report that read 'acid-fast bacilli (AFB) positive' or 'MTB detected' or 'strongly suggestive of TB' was considered as patients with TB. When in doubt, two physicians reviewed the reports and arrived at a consensus on the diagnosis. The PIS provided patient data for ATT purchased at the hospital's pharmacy.

Further, details of positive reports from sputum microscopy and culture registers were manually extracted and entered into Microsoft (MS) Excel as they were not available in the LIS.

A 'master database' for patients with TB diagnosed and/or treated in 2015 and 2016 was created using the unique hospital number (allocated to a patient at registration in the hospital) to match records and delete duplicate records in the various databases (PIS, LIS, RIS and manual registers).

A 'notification database' for patients with TB notified was also created. For this, data from the RNTCP register at the DOT centre of the hospital were entered into MS Excel. This was merged with data extracted from Nikshay portal. Patients diagnosed in late 2016 but who were notified in the first quarter of 2017 were also incorporated into this database.

In order to identify the proportions of TB cases notified, the 'master database' was matched with the 'notification database' using the VLOOKUP function in MS Excel. The patient's name was used as the primary matching variable. Records with a typographical mismatch in the patient's name were matched using a perfect match for 'sex' within an age range of ±3 years. Flow chart of data sources is depicted in figure 2.

### Qualitative component

We conducted 11 in-depth interviews (IDI) with various healthcare providers and one focus group discussion (FGD) with 11 nursing staff. At the time of the study, nursing staff looked after activities such as reporting of diseases, and we conducted an FGD with them as they comprised a fairly homogeneous group of female healthcare providers and were therefore included in an FGD. The first author (AS), a physician trained in qualitative research, conducted the interviews. Two of the interviews

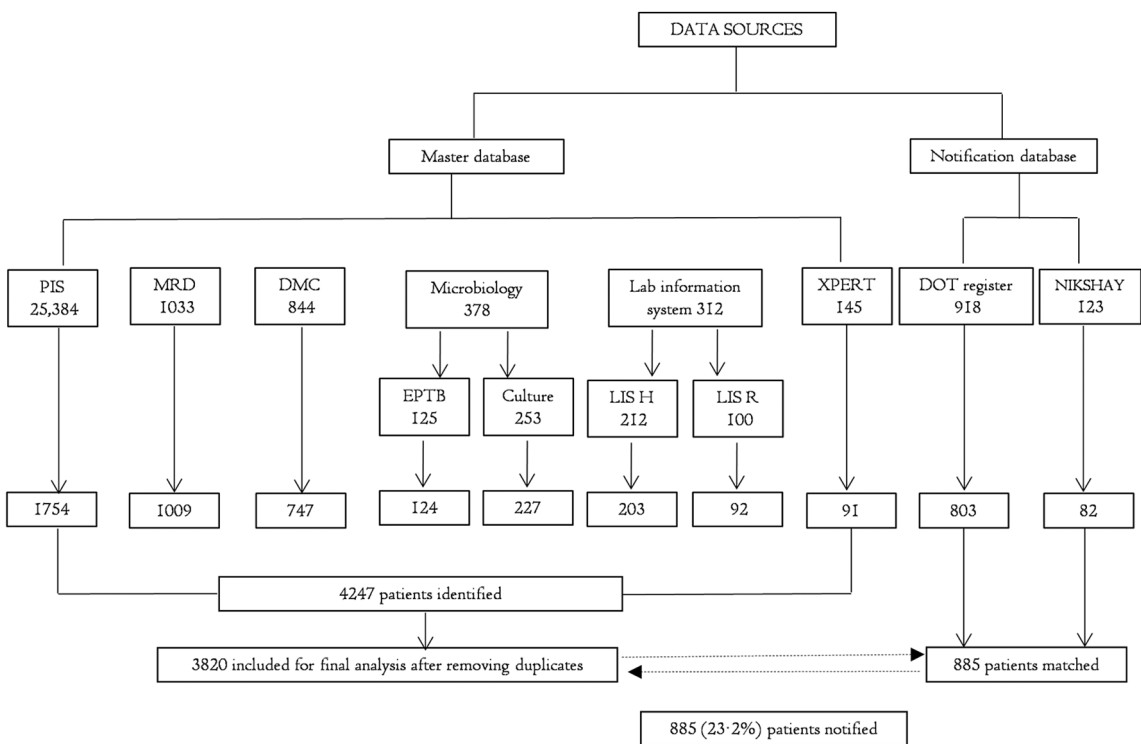

**Figure 2** Flow chart showing various data sources and proportion of tuberculosis (TB) notified to Revised National Tuberculosis Control Programme (RNTCP) from a private tertiary care teaching hospital in Bengaluru, India, 2015–2016. DMC, designated microscopy centre; DOT, direct observed treatment; EPTB, extrapulmonary tuberculosis register; LIS H, laboratory information system-histopathology; LIS R, laboratory information system-radiology; MRD, Medical Records Department; PIS, pharmacy information system; Xpert, GenXpert MTB/RIF.

were conducted in the local language, Kannada, and the rest in English. All interviews were audio recorded. A rapporteur made field notes during the interviews. After each interview, the key points were summarised and verified with the participants for validation. Data saturation guided the sample size. Each IDI lasted for 15–45 min and the FGD lasted for an hour.

### Data analysis

#### Quantitative component

EpiData (V.2.2.2.186, EpiData Association, Odense, Denmark) and Stata (V.12.1, Texas, USA) software were used for data analysis. The proportion of patients with TB notified was the outcome variable. Associations (unadjusted) between the outcome variable and demographic and clinical characteristics were derived using the $X^2$ test. All bivariate associations with a 'p value' <0.20 were included in a log-binomial regression model to obtain adjusted prevalence ratios with 95% CIs. A 'p value' <0.05 was considered statistically significant.

#### Qualitative component

All interviews and field notes were transcribed and translated into English for analysis using the 'thematic framework approach'. The first and last author (AS, RR) familiarised themselves with a few transcripts and manually coded them. The codes were then compared and categorised based on similarity. This formed the framework for the analysis.[17] The rest of the transcripts were subsequently indexed using the codes generated. Additional codes were added as and when necessary (box 1). The data were then summarised and mapped under various subthemes and themes which were reviewed by the rest of the authors for consensus.

### Patient and public involvement statement

Patients were not involved in the design or conduct of the study.

### Ethics

Ethics approval was obtained from the Institutional Ethics Committee, St John's Medical College, Bengaluru, Karnataka State, India, and the Ethics Advisory Group of the International Union Against Tuberculosis and Lung Disease, Paris, France. Permission to conduct the study and access hospital records was obtained from the chief of medical services at the hospital. Written informed consent was obtained from healthcare providers prior to interviews and included consent to audio record the interviews.

### RESULTS

#### Quantitative component

A total of 3820 patients were diagnosed with TB and/or treated during the study period. The demographic details of the patients with TB are described in table 1. The median (IQR) age was 40 (27–56) years and 7% of the patients were children <15 years of age.

---

**Box 1  Thematic framework used for understanding the issues with TB notification at a private tertiary care teaching hospital, Bengaluru, India, 2015–2016**

1.1. Gaps in the TB notification.1.1. Gaps in the TB notification.
1.2. Missing patients with TB.1.2. Missing patients with TB.
1.3. Confidentiality issue.1.3. Confidentiality issue.
2. Information to doctors.2. Information to doctors.
3. Disease disclosure to patients.3. Disease disclosure to patients.
4.1. TB diagnostic standard operating procedures.4.1. TB diagnostic standard operating procedures.
4.2. Technical issues associated with TB diagnosis.4.2. Technical issues associated with TB diagnosis.
5.1. Doctors' role in TB notification.5.1. Doctors' role in TB notification.
5.2. Reporting of patients with TB by doctors.5.2. Reporting of patients with TB by doctors.
5.3. Coordination between doctors and DOT centre.5.3. Coordination between doctors and DOT centre.
6.1. Standard operating procedures for TB notification.6.1. Standard operating procedures for TB notification.
6.2. Ease of notification.6.2. Ease of notification.
7.1. Policy decisions.7.1. Policy decisions.
8.1. Institute's notification policy.8.1. Institute's notification policy.
8.2. RNTCP notification policy.8.2. RNTCP notification policy.
8.3. Gaps in RNTCP notification policy.8.3. Gaps in RNTCP notification policy.
9.1. Streamlining TB notification.9.1. Streamlining TB notification.
9.2. Technological involvement.9.2. Technological involvement.

DOT, directly observed treatment short course; RNTCP, Revised National Tuberculosis Control Programme; TB, tuberculosis.

---

About a quarter of the patients received inpatient care and of them, nearly half were under the care of department of internal medicine, followed by chest medicine, neurology and paediatrics. About half of the patients with TB were identified through the pharmacy database while nearly 25% were identified through the LIS and laboratory registers.

Of the 3820 patients with TB, 885 (23.2%, 95% CI 21.9 to 24.5) were notified to the RNTCP. Of those notified, only 82 (9%) were also recorded in the Nikshay portal. Factors associated with notification are shown in table 2. Notification was significantly lower (unadjusted analysis) in children, inpatients and patients identified through the LIS and PIS. Notification was significantly higher for patients whose diagnosis was confirmed microbiologically (sputum smear microscopy, culture or GenXpert MTB/RIF). The final adjusted regression model showed age and sputum microscopy as determinants of notification.

#### Qualitative component

A total of 22 healthcare providers (11 from IDI and 11 from FGD) from various clinical departments at the hospital were interviewed. There were 10 physicians of whom seven were female. Six physicians had a work experience of >10 years. In addition, there were 12 paramedical staff including nurses, laboratory technicians and RNTCP staff most of whom had >10 years of work experience.

**Table 1** Demographic profile of patients with TB diagnosed and/or treated from 2015 to 2016 at a private tertiary care teaching hospital in Bengaluru, India

| Variable | n (%) | Notified (%) |
|---|---|---|
| Total | 3820 (100) | 885 (23.2) |
| Age (years) | | |
| 0–14 | 264 (6.9) | 24 (9.1) |
| 15–24 | 476 (12.5) | 118 (24.8) |
| 25–34 | 802 (21.0) | 166 (20.7) |
| 35–44 | 670 (17.5) | 159 (23.7) |
| 45–54 | 598 (15.7) | 160 (26.8) |
| 55–64 | 503 (13.2) | 129 (25.6) |
| 65 and above | 507 (13.3) | 129 (25.4) |
| Sex | | |
| Male | 2320 (60.7) | 559 (24.1) |
| Female | 1500 (39.3) | 326 (21.7) |
| Residence | | |
| Within state | 2362 (61.8) | 567 (24.0) |
| Outside state | 1358 (35.5) | 293 (21.6) |
| Not available | 100 (2.6) | 25 (25.0) |
| Marital status | | |
| Unmarried | 1008 (26.4) | 183 (18.2) |
| Married | 2604 (68.2) | 653 (25.1) |
| Others | 208 (5.4) | 49 (23.6) |
| Year diagnosed | | |
| 2015 | 2071 (54.2) | 482 (23.3) |
| 2016 | 1749 (45.8) | 403 (23.0) |
| Inpatient | | |
| Yes | 1009 (26.4) | 137 (13.6) |
| No | 2811 (73.6) | 748 (26.6) |
| Department (n=1009) | | |
| Medicine | 484 (48.0) | 64 (13.2) |
| Pulmonary medicine | 141 (14.0) | 21 (14.9) |
| Paediatrics | 81 (8.0) | 16 (19.8) |
| Neurology | 88 (8.7) | 5 (5.7) |
| General surgery | 41 (4.1) | 5 (12.2) |
| Orthopaedics | 50 (5.0) | 8 (16.0) |
| Others | 124 (12.3) | 15 (12.1) |
| Source of patients with TB* | | |
| Sputum microscopy register | 747 (19.6) | 481 (64.4) |
| Extrapulmonary TB positive register | 124 (3.2) | 24 (19.4) |
| Histopathology database | 203 (5.3) | 53 (26.1) |
| Radiology database | 92 (2.5) | 13 (13.7) |
| Pharmacy database | 1754 (45.9) | 341 (19.4) |

Continued

**Table 1** Continued

| Variable | n (%) | Notified (%) |
|---|---|---|
| Culture register | 227 (5.9) | 72 (31.7) |
| GenXpert MTB/RIF register | 91 (2.4) | 38 (41.8) |
| Inpatient database | 1009 (26.4) | 137 (13.6) |

*Cumulative percentage may add up to more than 100 since one patient could have tested positive by more than one diagnostic method.
TB, tuberculosis;

The four themes that emerged through the qualitative analysis were: (1) basic diagnostic modalities and treatment promote notification of TB; (2) misconceptions regarding notification and its process are common among healthcare providers; (3) despite a national notification system other factors prevented notification of all patients; and (4) establishing hospital systems for notification will go a long way in improving notifications (tables 3 and 4).

### Basic diagnostic modalities and treatment promote notification of TB

Patients whose diagnosis was based on sputum microscopy and those receiving treatment through the RNTCP were more likely to be notified than those requiring complex diagnostics.

#### *Patients who are sputum positive for TB bacteria are more likely to be notified*

Diagnosis based on simple sputum smear microscopy was more likely to lead to notifications than patients requiring complex diagnostics such as radiography, biopsies, tissue examinations, bacteriological cultures or non-traditional laboratory diagnostics such as GenXpert MTB/RIF and irrespective of whether these were inpatients or outpatients.

It was perceived that the RNTCP guidelines for notification restrict notification to only those patients diagnosed with MDR TB at an RNTCP accredited laboratory. Hence, patients with MDR TB were not notified.

> Confirmation from [an Intermediate Reference Laboratory (IRL)] is a must for initiating the MDR regimen, without this MDR TB patients cannot be (treated with DOT) or notified. (Paramedical staff 9 (IDI))

#### *Notifications are more likely for those diagnosed with pulmonary TB*

Most referrals to the RNTCP DOT centre were of patients diagnosed with pulmonary TB. Most patients with extrapulmonary TB were prescribed ATT through the hospital's pharmacy and therefore bypassed the DOT centre and hence notification.

**Table 2** Factors associated with TB notification at a private tertiary care teaching hospital in Bengaluru, India, from 2015 to 2016

| Variable | Total | Notification n (%) | Crude PR (95% CI) | P value | Adjusted PR (95% CI) | P value |
|---|---|---|---|---|---|---|
| Total | 3820 | 885 (23.2) | – | – | – | – |
| Age (years) | – | – | – | – | – | – |
| Children (<15) | 264 | 24 (9.1) | 1 | – | 1 | – |
| Adults (≥15) | 3556 | 861 (24.2) | 2.6 (1.8 to 3.9) | 0.000* | 1.5 (1.0 to 2.2) | 0.039* |
| Sex | – | – | – | – | – | – |
| Female | 1500 | 326 (21.7) | 1 | – | – | – |
| Male | 2320 | 559 (24.1) | 1.1 (0.9 to 1.2) | – | – | – |
| Marital status | – | – | – | – | – | – |
| Unmarried | 1008 | 183 (18.2) | 1 | – | 1 | – |
| Married | 2604 | 653 (25.1) | 1.3 (1.1 to 1.5) | 0.000* | 1.0 (0.9 to 1.2) | 0.240 |
| Others | 208 | 49 (23.6) | 1.2 (0.9 to 1.7) | 0.066 | 1.1 (0.8 to 1.4) | 0.346 |
| Inpatient | – | – | – | – | – | – |
| No | 2811 | 748 (26.6) | 1 | – | 1 | – |
| Yes | 1009 | 137 (13.6) | 0.4 (0.4 to 0.5) | 0.000* | 1.0 (0.8 to 1.2) | 0.925 |
| Residence | – | – | – | – | – | – |
| Within state | 2362 | 567 (24.0) | 1 | | – | |
| Outside state | 1358 | 293 (21.6) | 0.8 (0.7 to 1.0) | 0.092 | – | – |
| Not recorded | 100 | 25 (25.0) | 1.0 (0.7 to 1.4) | 0.819 | – | – |
| Year diagnosed | – | – | – | – | – | – |
| 2015 | 2071 | 482 (23.3) | 1 | – | – | – |
| 2016 | 1749 | 403 (23.0) | 0.9 (0.8 to 1.1) | 0.866 | – | – |
| Sputum smear microscopy | – | – | – | – | – | – |
| Positive | 747 | 481 (64.4) | 4.8 (4.4 to 5.4) | 0.000* | 4.7 (4.1 to 5.3) | 0.000* |
| Others | 3073 | 404 (13.1) | 1 | – | 1 | – |
| EPTB microscopy register | – | – | – | – | – | – |
| Positive | 124 | 24 (19.4) | 0.8 (0.5 to 1.1) | 0.318 | – | – |
| Others | 3696 | 861 (23.3) | 1 | – | – | – |
| Culture | – | – | – | – | – | – |
| Positive | 227 | 72 (31.7) | 1.4 (1.1 to 1.7) | 0.001* | 1.0 (0.8 to 1.2) | 0.855 |
| Others | 3593 | 813 (22.6) | 1 | | 1 | |
| GenXpert MTB/RIF | – | – | – | – | – | – |
| Positive | 91 | 38 (41.8) | 1.8 (1.4 to 2.3) | 0.000* | 1.1 (0.9 to 1.3) | 0.295 |
| Others | 3729 | 847 (22.7) | 1 | – | 1 | – |
| Histopathology database | – | – | – | – | – | – |
| Present | 203 | 53 (26.1) | 1.1 (0.8 to 1.4) | 0.299 | – | – |
| Others | 3617 | 832 (23.0) | 1 | – | – | – |
| Radiology database | – | – | – | – | – | – |
| Present | 92 | 13 (13.7) | 0.5 (0.3 to 0.9) | 0.038* | 0.7 (0.4 to 1.2) | 0.285 |
| Others | 3725 | 872 (23.4) | 1 | – | 1 | – |
| Pharmacy database | – | – | – | – | – | – |
| Present | 1754 | 341 (19.4) | 0.7 (0.6 to 0.8) | 0.000* | 0.9 (0.8 to 1.0) | 0.839 |
| Others | 2066 | 544 (26.3) | 1 | | 1 | |

*Significant p value.
EPTB, extrapulmonary tuberculosis; PR, prevalence ratio; TB, tuberculosis.

**Table 3** Barriers and solutions identified for TB notification at a private tertiary care teaching hospital in Bengaluru, India, 2015–2016

| Barriers | Solutions |
| --- | --- |
| Patients with TB diagnosed by culture, histopathology, radiology, BAL usually missed | Integration of LIS |
| Incomplete notification among inpatients | Triangulation of TB data from all possible sources |
| MDR TB missed | Proper documentation and communication which helps in notification |
| Lack of dedicated manpower | Appointment of notification officer |
| Non-DOT not notified | Referral of all patients started on ATT by the treating doctor to the notification officer |
| Knowledge issues | Awareness about notification communicated |
| Lack of capacity building | Refresher trainings about Nikshay |
| Absence of hospital notification policy and standard operating procedure | Institutional notification policy |
| Inadequate networking between stakeholders | Having single notification desk with dedicated telephone number |
| Patient confidentiality concerns | Patient counselling about the importance of notification, ensuring adequate cyber security |
| Duplication of data | Unique identifier (such as social security number, Aadhaar number in India) to prevent duplication that helps notify, track and retain patient in care |

ATT, anti-TB treatment; BAL, bronchoalveolar lavage; DOT, directly observed treatment short course; LIS, laboratory information system; MDR TB, multidrug-resistant tuberculosis.

Almost 85% extra-pulmonary patients don't take DOT or don't go to TBHV (who in turn notifies). (Paramedical staff 1 (IDI))

*Receiving treatment through the RNTCP is synonymous with notification*

Not all patients are initiated on DOT through the RNTCP. Some are prescribed ATT through the hospital's pharmacy at their own expense. As the responsibility for notification lies with the DOT centre, patients not referred to the DOT centre are not notified. Few medical and paramedical personnel knew the procedure for notification of TB at the hospital.

The physician that reviews the patients (reports do not advise DOT) so the RNTCP staff is not aware (of the patient diagnosed with TB). At least if the patients visit the DOT center, (the RNTCP staff) will know… but 50% of the patients treated by doctor are not referred to the DOT center. (Paramedical staff 1 (IDI))

Whoever goes to the DOT center (gets) registered and notified. (Physician 1 (IDI))

### Misconceptions regarding notification and its process are common among healthcare providers

The level of knowledge and awareness regarding notification and its systems was poor. Healthcare providers did not perceive notification as their responsibility.

*Those who do not know, do not notify: awareness could improve notification*

Some healthcare providers were unaware that TB was a notifiable disease, others were unsure of the existing system for notifying TB and yet others presumed that notification was common knowledge. Out of 22 healthcare providers, 14 were aware of the RNTCP requirement of notification.

I don't think TB is a notifiable disease, is it a notifiable disease? That means every TB patient we come across (should be) notified? And to whom should we notify? (Physician 10 (IDI))

*Notification is someone else's responsibility*

There was confusion regarding the responsibility for notification. Many healthcare providers considered notification the responsibility of the RNTCP and not of the institution. The laboratories considered notification the responsibility of the treating physician and vice versa.

What we assume is that, the patient will go back to the doctor, maybe the doctor has to notify it. (Paramedical staff 5 (FGD))

I think from the labs they notify directly, we haven't taken it on us to notify as yet. (Physician 2 (IDI))

### Despite a national notification system, other factors prevent notification of all patients

Inadequate training for using the notification portal Nikshay and mandatory information requirements within the portal were barriers to notification.

*Inadequate user training interferes with notifications via Nikshay portal*

There were mixed opinions regarding notification via Nikshay portal. While some considered Nikshay easy to use, few remembered having any training to use Nikshay for notification. Regular updates within the Nikshay portal without training to handle updates also interfered with notifications.

Nikshay is (quite) easy, what we had seen during the Nikshay demo, seemed okay. (Physician 2 (IDI))

**Table 4** A brief description of the framework used in the qualitative data analysis for understanding TB notification at a private tertiary care teaching hospital in Bengaluru, India, 2015–2016

**Thematic framework components and quotes**

| Thematic framework components and quotes | Codes | Summary | Categories | Subthemes | Themes* |
|---|---|---|---|---|---|
| *Technical issues associated with TB diagnosis:* The GenXpert MTB/RIF lab here hasn't received accreditation. So we can't take that, even if it's positive. Since no accreditation, no treatment can be given from government. | Perception that confirmation from IRL is a must for putting on MDR regimen, without which such patients cannot be notified and hence will be put on non-DOT. | Inability of RNTCP staff to notify patients positive for MDR by GenXpert MTB/RIF due to unclear instructions related to RNTCP accreditation of laboratory. | MDR TB not notified due to unclear instructions. | Quality issues interfere with multidrug-resistant TB notification. | 1 |
| *TB notification standard operating procedures:* Because none of us know when to notify and how to notify, I may not have notified. | Standard operating procedures involved in notification unknown. | Lack of knowledge about the process of notification and assuming somebody else has to notify. | Lack of knowledge regarding notification. | Notification is someone else's responsibility. | 2 |
| *Gaps in RNTCP notification policy:* There are additions in NIKSHAY, still they haven't given us proper training. | Gaps in notification. | Refresher training on Nikshay has not been given to the RNTCP staff involved in notification even when new forms have been updated in the software. | Lack of basic training in Nikshay. | Gaps in user training for the notification portal Nikshay. | 3 |
| *Technological involvement:* The moment we have electronic medical records, if anyone is given ATT and is done online…we would get much better way of tracking them. | Triangulation of patients diagnosed or treated from all departments. | Scope for integrating electronic medical records with case diagnosis which will ease notification. | Technical solutions to improve notification. | Record linkage through unique identification numbers. | 4 |

*Theme 1—traditional diagnostic procedures promote notification of patients with TB. Theme 2—misconceptions regarding notification and its process are common in healthcare providers. Theme 3—despite a national notification system, other factors prevented notification of all patients. Theme 4—establishing hospital systems for notification will go a long way in improving notifications.

ATT, anti-TB treatment; DOT, directly observed treatment short course; IRL, Intermediate Reference Laboratory; MDR TB, multidrug-resistant tuberculosis; RNTCP, Revised National Tuberculosis Control Programme.

There are changes that are made to the Nikshay portal… they haven't trained us adequately for it. (Paramedical staff 10 (IDI))

### Fear of compromising privacy interferes with notification

Fear of stigma from a breach in confidentiality prevents patients from sharing personal identifiers such as phone numbers. This limits entries into the notification portal due to missing information in 'mandatory fields'.

### Establishing hospital systems for notification will go a long way in improving notifications

Notification policy, standard operating procedures and dedicated personnel supported with innovative technologies such as hot lines and mobile applications were suggested.

### Comprehensive institutional notification policy for TB: a necessity

Developing and implementing a comprehensive institutional notification policy to improve notification was suggested. This policy was expected to provide guidance for delegating responsibilities and linking the various components of the hospital information system to enable identification and notification of patients with TB.

### Dedicated human resources could bridge gaps in the existing notification system

Supplementing the existing human resource for notification, that is, TBHV and LT, with a dedicated notification officer (institutional) and an RNTCP medical officer at the DOT centre (via the programme) who could liaison with each other was considered essential.

Let's say, we appoint a person with an intercom or maybe a mobile (phone) so that the physician just calls that person and (informs)…then s/he could probably follow-up the patient to (obtain) the details…. (Physician 9 (IDI))

### Linking records through a unique identification number is useful

Documenting the hospital number in the RNTCP register and the government-issued unique identification number (Aadhaar number)[18] in TB notification portal Nikshay could enable linkage while preventing duplication.

### Developing innovative information, communication and technology (ICT) support systems to aid notification

A 'one window' concept, that is, establishing a dedicated notification hot line, or a mobile phone application to feed the details of patients with TB diagnosed and treated at the hospital, was suggested. Developing algorithms to short-list those diagnosed with TB from the LIS, along with electronic linking of outpatient, inpatient, laboratory, diagnostic and pharmacy records, was considered to support universal notification.

We have to go electronic and we have to then integrate everything…ordering (drug prescription) online…the moment we have electronic medical records… we would get much better way of tracking them. (Physician 7 (IDI))

## DISCUSSION

Indian private healthcare sector contributes to only one-fifth of the TB notification in the country.[9] Few reports have explored existing gaps in notification within the private sector. To our knowledge, this is the first report on the extent of TB notification and its challenges from a private tertiary care teaching hospital in India.

As in other studies, poor awareness and attitudes along with inadequate systems limited the TB notifications at the hospital to a quarter of those diagnosed.[19–21] Some private practitioners are of the opinion that notification of TB is unlikely to bring about change in prescription practices and question the need for collecting personal information that does not lead to public health action.[14] Therefore, training and sensitisation of healthcare personnel for notification is recommended. Such training should focus on the benefits of notification from the public health and ethical perspective.[13] It is also essential for the RNTCP to provide annual feedback to healthcare providers of the numbers notified and how this affects policy for TB care. Additionally, obtaining feedback from private practitioners regarding the notification process is expected to boost provider's morale and thereby notifications.[11]

Linking hospital records electronically could simplify notification. This does not eliminate manual data entry into the notification portal. Software solutions that feed data to the notification portal automatically could simplify notification and are currently being explored for MDR diagnostic machines.[8] Further, applying ICD-10 codes for diagnoses, commonly used within TB notification systems globally,[11] could standardise diagnoses, enable data capture through software systems and simplify notification.

The guidance for TB notification in India suggests the appointment of a TB nodal officer.[8] The TBHV who currently fulfils this role in our context is probably overburdened with responsibilities in the absence of the 'DOT centre medical officer,' a functionary of the RNTCP. Reports from the private sector also indicate the need for additional human resources in the light of the volume of patients that they cater to.[22] Identifying an additional 'nodal officer' for TB notification from among existing institutional personnel could optimise the use of existing resources for notification.

Healthcare providers suggested innovative ICTs such as mobile applications for notification. However, the short messaging service, interactive voice response or phone calls to notify TB enabled by the RNTCP for notification are not as popular as expected. Further, though, the Nikshay mobile application that is underway to simplify the notification process holds promise,[23] whose effectiveness remains to be explored.

Healthcare providers elsewhere in India recommend simplifying the existing notification technology to promote uptake.[23] Regular training that includes Nikshay updates was widely requested, but currently negligible could remove existing technological barriers and enhance notification.

As the DOT centre at the hospital is located within the Chest Medicine Department, it is not surprising that sputum-positive patients are notified. Only 17% of chest physicians notified TB, reflecting the gap between awareness and action.[12] However, in our study, ownership of the DOT centre probably made notification a responsibility of the chest physician and enhanced their engagement with the RNTCP. Locating DOT centres within clinical departments with the largest burden of patients with TB to improve notifications is worth exploring.

Though the Indian Academy of Paediatrics supports TB treatment through the RNTCP,[24] the proportion of children with TB notified was low, reflecting the limited involvement of paediatricians in the RNTCP. The questionable bioavailability of paediatric ATT formulations and alternate day dosing schedules are known barriers to engaging paediatricians with the RNTCP.[25] The introduction of the daily regimen with 'body weight bands' that inform dosing has the potential to improve provider engagement with the RNTCP and improve TB notifications thereof, irrespective of the patients' age.[16] Also, creating a TB registry within each clinical department could improve departmentwise notifications.

Linking patient records using a unique identification number (Aadhaar number)[18] or hospital number, and extending this system to involve the Nikshay portal could minimise duplication, simplify record and help retain patients in care. Studies indicate that patients are weary of sharing personal identifiers, that is, mobile phone numbers and address, for notification.[10 12] This necessitates sensitising the general public of the need for mandatory disease notification through mass media campaigns and patients through counselling sessions. Further, perceived stigma prevents healthcare providers involved in notification uncomfortable with obtaining personal identifiers from patients.[10 12] Mobile phone numbers and the patient's address are mandatory fields in the Nikshay portal, without which notification is incomplete. Therefore, reminding healthcare providers of their obligation to obtain and report personal identifiers of patients with notifiable disease, as per the Indian Medical Council's Regulations 2002,[26] might minimise discomfort in the light of responsibility. Simultaneously, mass media, posters and brochures placed in waiting rooms regarding notification could mitigate patients' fears with sharing personal identifiers.

Though punitive action for non-notification exists in India, it is not yet implemented.[26] As in other TB high-burden countries, a recent mandate suggests that non-notification could result in heavy fines and even imprisonment.[27] In the light of Government of India's politicoadministrative commitment towards TB control,

punitive action is an eventuality that is best avoided. Therefore, at institutional level, enabling incentives for notification (tangible or intangible) and disincentives for non-notification ('warnings/memos', or monetary penalty) could reinforce the importance of notification. Further, the RNTCP provides a cash incentive of 250 INR to a 'private' healthcare provider for every patient with TB notified.[28] Institutional proactiveness to ensure that its healthcare providers receive this incentive could also improve notifications.

Finally, testing for MDR TB cases in the study hospital was done using GenXpert MTB/RIF equipment that was acquired through the Initiative for Promoting Affordable and Quality TB Tests (IPAQT) project. IPAQT aims to bring WHO-approved TB tests at affordable prices to patients in private sector.[29 30] This was the only MDR TB diagnostic service available at the hospital. MDR TB cases detected through IPAQT are entered into Nikshay through a subportal within the hospital's primary Nikshay portal. It is only through this subportal that a person diagnosed with MDR TB at the hospital could be notified. A lack of awareness of this separate portal prevented notification of MDR TB diagnosed in the hospital. Understanding these issues at project initiation, documenting project procedures and ensuring 'complete knowledge transfer' to institutional personnel when institutions absorb such projects is necessary.

### Methodological issues

The mixed-methods design with the quantitative and qualitative components validated and complemented each other. It is possible that our definition of notification overestimated the numbers notified. We were also liberal with our criteria for matching databases. However, we included all patients both diagnosed and treated at the hospital even if they availed a 'one-time' consultation. This probably also inflated the denominator minimising any overestimate.

The retrospective nature of the quantitative component meant that the study procedures did not influence changes in notification, as might have been observed if the study were prospective. Further, the quantitative component, based on a review of records, is limited by the quality of the data in the records, for example, we could not assess the association of the treating clinical department and treatment regimens on notification.

The study included healthcare providers who encountered patients with TB at different points in the hospital as represented in figure 1, including hospital staff and RNTCP staff. Therefore, we believe that this sample is fairly representative of those healthcare providers who manage patients with TB. This, along with a description of the study context and methodology, enables the reader to judge its applicability of the results to their context. The first author's position as a physician in the study setting helped her contextualise the results. Sharing the results with all authors with diverse backgrounds and skills improved the interpretation of the results, further improving generalisability.

## CONCLUSIONS

The low proportions of TB notifications at the hospital call for urgent action to identify strategies that can improve notification. A combined approach from within (managerial) and outside the institution (RNTCP) is necessary. Generating awareness regarding notification and developing a comprehensive notification policy along with establishing a notification portal is essential. Supplementing this with technological innovations such as mobile applications and expanding the scope of the existing hospital information system to capture outpatient data and link patient records is essential.

We also call on tertiary-level teaching hospitals both within India and globally to evaluate the TB notifications and its barriers in their setting. Such information is hoped to support the development of evidence-based strategies that enhance public-private engagement for TB notification and control.

**Author affiliations**
[1]Department of Community Health, St John's National Academy of Health Sciences, Bengaluru, Karnataka, India
[2]St John's Research Institute, St John's National Academy of Health Sciences, Bengaluru, Karnataka, India
[3]Operational Research, International Union Against Tuberculosis and Lung Disease, Paris, France
[4]Operational Research, International Union Against Tuberculosis and Lung Disease, New Delhi, India
[5]Department of Pulmonary Medicine, St John's National Academy of Health Sciences, Bengaluru, Karnataka, India
[6]Institute of Medicine, University of Chester, Chester, UK
[7]Department of Medical Research, Ministry of Health and Sports, Yangon, Myanmar
[8]Department of Public Health Sciences, Karolinska Institute, Stockholm, Sweden
[9]Wellcome Trust/DBT India Alliance, Banjara Hills, Hyderabad, India

**Acknowledgements** This research was conducted through the Structured Operational Research and Training Initiative (SORT IT), a global partnership led by the Special Program for Research and Training in Tropical Diseases at the World Health Organization (WHO/TDR). The model is based on a course developed jointly by the International Union Against Tuberculosis and Lung Disease (The Union) and Medécins sans Frontières (MSF/Doctors Without Borders). The specific SORT IT programme which resulted in this publication was jointly developed and implemented by: The Union South-East Asia Office, New Delhi, India; the Center for Operational Research, The Union, Paris, France; The Union, Mandalay, Myanmar; MSF Luxembourg Operational Research (LuxOR); MSF Operational Center Brussels (MSF OCB); Institute of Medicine, University of Chester, UK; and Department of Medical Research, Ministry of Health and Sports, The Republic of The Union of Myanmar.

**Contributors** AS, GDS and RR conceived the study. AMVK and EW along with AS, GDS and RR designed the protocol. MNA did the literature search. AS and MNA did data entry and extracted the electronic data. AS undertook the interviews. AS, AMVK, RR and TMM analysed and interpreted the data. AS drafted the manuscript with support from AMVK and RR. GDS and EW critically revised the successive drafts. All authors have seen and approved the final version of this manuscript for publication.

**Funding** The training programme and open access publications costs were funded by the Department for International Development (DFID), UK and La FondationVeuve Emile Metz-Tesch (Luxembourg). The study is supported by grants from the Swedish Research Council and the Wellcome Trust/DBT India Alliance.

**Disclaimer** The funders had no role in study design, data collection and analysis, decision to publish, or preparation of the manuscript. The corresponding author (AS) confirms that she had full access to all the data in the study and had final responsibility for the decision to submit for publication.

**Competing interests** AS, GDS and RR are employed at the tertiary-level teaching facility that was evaluated in this study.

**Patient consent** Not required.

**Ethics approval** Institutional Ethics Committee, St John's Medical College and Hospital, and Ethics Advisory Group of the International Union Against Tuberculosis and Lung Disease, Paris, France.

**Provenance and peer review** Not commissioned; externally peer reviewed.

**Data sharing statement** Data will be made available with reasonable request from the corresponding author.

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
