## [Reviewer comments · BMJ Open]

This paper was submitted to a another journal from BMJ but declined for publication following peer review. The authors addressed the reviewers' comments and submitted the revised paper to BMJ Open. The paper was subsequently accepted for publication at BMJ Open.

(This paper received three reviews from its previous journal but only two reviewers agreed to published their review.)

ARTICLE DETAILS

TITLE (PROVISIONAL)	Tuberculosis Notification in a Private Tertiary Care Teaching Hospital in South India: a Mixed Methods Study
AUTHORS	Siddaiah, Archana; Ahmed, Naseer; Kumar, Ajay; D'Souza, George; Wilkinson, Ewan; Maung, Thae Maung; Rodrigues, Rashmi

VERSION 1 – REVIEW

REVIEWER	Premila Webster Nuffield Department of Population Health, University of Oxford UK
REVIEW RETURNED	30-May-2018

GENERAL COMMENTS	This is an important area of research where there appears to be a paucity of work specifically in this area. While a significant proportion of TB is diagnosed and treated in the private sector the notification of TB from private hospitals is limited. This will result in issues related to accurate data to enable appropriate monitoring, evaluation of treatment, etc. This study systematically explores the proportion not notified from a private hospital and the reasons for this under notification and possible solutions through the qualitative component. The research question is clear, methods used and analysis appropriate and conclusions reached justified. The paper is clearly written and structured. There are a few typos and minor grammatical errors. The limitations are clearly outlined, A useful and good paper overall, which would inform and benefit those involved both in research and policy in this area.
--

REVIEWER	Sally-Ann Ohene WHO Country Office, Ghana
REVIEW RETURNED	13-Jul-2018

GENERAL COMMENTS	GENERAL REMARKS This study highlights the status of TB notification from a tertiary private hospital and the barriers to notification. It is an interesting study generally well written with findings worth publishing. There are one or two grammatical errors that need correction. A clearer version of the font in Figures 1 and 2 would make them easier to read. Some suggestions are provided below to improve the paper.
---

ABSTRACT

The abstract gives an appropriate summary of the study and findings. In the Results, there is a need to revise the statement “Despite efforts Nikshay does not ensure notification of all patients with TB” so that it is clearly understood by readers. It is unclear as it stands.

For one, we are not familiar with Nikshay. Secondly, the statement itself does not provide enough information on why notification of TB patients is not ensured.

In the conclusion, the statement “Combined approach from within (managerial) and outside the institution i.e. Revised National Tuberculosis Control Programme (RNTCP) is necessary” appears incomplete. Please complete the sentence to indicate what the combined approach is necessary for.

INTRODUCTION

Great introduction.

Page 5 Line 17 It would be useful to include a statement of the proportion of TB patients that access care in the private sector and the estimated rate of notification to the NTP if such data is available.

METHODS

It would be useful to provide a brief information of TB control program activities in India for readers who might not be familiar with the set up.

Please provide the RNTCP’s definition of a TB patient. How does it compare with the study’s definition of a TB case?

Please clarify if the definition of the TB patient was taken as the inclusion criteria for the study.

What were the exclusion criteria?

Page 8 Line 15: In reference to patient details, please indicate the type of data variables that were extracted for the study participants.

Page 9 Line 33: Please indicate how people were in the focus group discussion (FGD) and the rationale for involving only nurses in the FGD.

RESULTS

In the methods, it is indicated that 11 in-depth interviews were conducted. Yet in the results it is stated that 22 health care staff were interviewed. Please resolve the discrepancy.

Page 17 Line 40 Please provide information on the proportion of those interviewed who were aware of the RNTCP P requirement of notification.

Please explain whether for the qualitative results, the statements in italics are statement from the focus groups or in-depth interviews.

It is not quite clear which is which as some have what appears to be the informant stated which others are just presented.

Please organize the results in a way that indicates what emerges from the informant interviews and what is from the FGD.

DISCUSSION

The discussion highlights important information and recommendations for addressing the gaps in TB notification identified in the study.

	I am also expecting to find how the study findings compared with what other studies in India and elsewhere have found. Please address this gap in the discussion. Page 22 Lines 5 to 18 I was expecting to see this information in the introduction to put the study within the context of the prevailing situation. I suggest that this section is moved to the Introduction. Page 22 Line 31. Please provide the references Page 25 Line 44 to Page 26 Line 12 is not clear. What project is being referred? How is this project related to the study. This section is a bit confusing and should be clarified. Page 26 Line 31 Sentence is not clear. Please rephrase. Page 26 Line 43 The informants were said to have been conveniently; therefore to now indicate that those interviewed were fairly representative of the health care staff warrants some more information in the Results section on how those interviewed reflects the population of health staff. CONCLUSION The conclusion is in line with the study findings.
--	--

VERSION 1 – AUTHOR RESPONSE

GENERAL REMARKS

This study highlights the status of TB notification from a tertiary private hospital and the barriers to notification. It is an interesting study generally well written with findings worth publishing. There are one or two grammatical errors that need correction. A clearer version of the font in Figures 1 and 2 would make them easier to read.

Authors' Response: *Thanks for this comment. We have read through the paper again and corrected the grammatical errors and fonts.*

Some suggestions are provided below to improve the paper so that it is clearly understood by readers.

ABSTRACT

The abstract gives an appropriate summary of the study and findings. In the Results, there is a need to revise the statement "Despite efforts Nikshay does not ensure notification of all patients with TB" It is unclear as it stands. For one, we are not familiar with Nikshay. Secondly, the statement itself does not provide enough information on why notification of TB patients is not ensured.

Authors' Response: *Thank you for this comment. We have rephrased the sentence and clarified what Nikshay is in the abstract as follows:*

Abstract: From the year 2012, India's TB notification mandatory and launched Nikshay case-based web portal to facilitate notification from both the public and private sector.

Manuscript introduction: Page 5 paragraph 2: This despite launching Nikshay, the case based web based national TB notification portal, accessible to all healthcare providers, laboratories and diagnostic facilities, both public and private, nationwide.

Page 5 paragraph 2: clarifies why notification of TB patients is not ensured: Improving the estimates of disease prevalence through are essential to planning, monitoring, and evaluation of the RNTCP. Yet, barriers such as lack of time, low awareness regarding notification, concern towards breaching patient confidentiality, operational complexities in notifying along with lack of trust in the public sector contribute to poor TB notifications from the private sector. [13–15]

The theme emerging out of qualitative data has also been rephrased for more clarity into “despite a national notification system, other factors have prevented notification of all patients”.

Question:In the conclusion, the statement “Combined approach from within (managerial) and outside the institution i.e. Revised National Tuberculosis Control Programme (RNTCP) is necessary” appears incomplete. Please complete the sentence to indicate what the combined approach is necessary for.

Authors’ Response:*Thank you very much!We have now rephrased and completed the statement as follows:*

A comprehensive approach both by the hospital management and the national TB programme is necessary for improving notification. Such a comprehensive notification policy include, improving awareness among health care providers about the requirement for TB notifications, establishing a single notification portal in-hospital, digitally linking hospital records to Nikshay and designating a notification specific human resource.

Reviewer 2:

INTRODUCTION

Great introduction.

Authors’ Response:*Many thanks!!*

Question:Page 5 Line 17 It would be useful to include a statement of the proportion of TB patients that access care in the private sector and the estimated rate of notification to the NTP if such data is available.

Authors’ Response:*Thank you for the suggestion. We agree that this information is crucial to know the gaps in TB notification across health systems. Therefore we have inserted the proportions notified from public and private health systems as presented below.*

Page 5 paragraph 2:

Healthcare delivery in India involves both the public and private sectors. The Indian private healthcare sector is estimated to cater to approximately 2/3rd of the inpatients and 3/4th of the outpatients in the country.[16] The private healthcare sector also accounts for 54% of the healthcare teaching facilities in India.[16]It is therefore not surprising that approximately 2/3rd of the 2.2 million patients with TB annually are diagnosed and treated within the private healthcare sector.[5] However, in 2017 only 19% (81% from public sector) of these patients receive care from, or are notified i.e., reported, to the Revised National Tuberculosis Control Programme (RNTCP)[4,6], India’s national health program for the prevention and control of TB. Though, mandatory TB notification introduced in 2012 saw a sharp increase in TB notifications, notification from the private sector continues to below.[4,6–10]. This despite launching Nikshay, the case based web based national TB notification portal, accessible to all healthcare providers, laboratories and diagnostic facilities, both public and private, nationwide.

METHODS

Question:It would be useful to provide a brief information of TB control program activities in India for readers who might not be familiar with the set up.

Authors’ Response:*A brief introduction has now been provided as follows **on page 7** of the manuscript:*

The Indian RNTCP and its relationship with the study hospital:

Administrative setup:

The RNTCP, a vertical national health program, strives to provide care and treatment at no cost to all patients with TB in India. The program adheres to the diagnostic and treatment recommendations of the World Health Organization (WHO). The RNTCP delivers its services through designated microscopy

center (DMC, population covered: 0.1 million)>peripheral health institutions (PHI, primary, secondary and tertiary healthcare facilities including all healthcare academia).

In addition, Direct observed treatment (DOT) centers at PHIs are responsible for dispensing treatment, observing treatment doses swallowed (DOT), patient follow-up and patient retention in care. Till 2017, the RNTCP followed an alternate treatment regimen, with DOT thrice a week in the intensive phase (2 months) and weekly once in the continuation phase (4 months). All public PHIs function as DOT centers and have a TB health visitor (TBHV), responsible for DOT and patient retention. DOT centers at academic institutions however, have a medical officer in addition to the TBHV. A PHI may also function as DMC.

Private sector: The RNTCP sets guidelines but does not dictate treatment protocols to the private sector. In addition it attempts to deliver public services to private sector through public private partnerships (PPP). However, all private healthcare providers are expected to notify TB patients irrespective of a PPP through Nikshay.

Question:Please provide the RNTCP's definition of a TB patient. How does it compare with the study's definition of a TB case?

Authors' Response:*The RNTCP definition of a TB patient has been given below.*

RNTCP has two definitions- a) microbiologically confirmed TB case- refers to a presumptive TB patient with biological specimen positive by any of the diagnostic methods (microscopy, culture, molecular testing)& b) Clinically diagnosed TB case-refers to a presumptive TB patient who is not microbiologically confirmed, but diagnosed with active TB by a clinician.

Even though RNTCP definitions are not mentioned in the manuscript, we have mentioned that both the definitions are similar and our study definition is in line with the RNTCP definition as mentioned below.

We have clarified this as follows on page 8 paragraph 3.

For this study, the definition of a patient with TB incorporated the RNTCP definitions and included pharmacy records a surrogate for those diagnosed as outpatients in absence of an electronic health record for outpatients. A patient with TB was therefore defined as (i) Microbiologically confirmed (RNTCP): a patient with microbiologically confirmed TB using microscopy, bacterial culture, and/or GenXpert MTB/RIF® OR (ii) Clinically diagnosed (RNTCP): a patient with histopathological or radiological findings suggestive of TB, irrespective of microbiological confirmation, OR (iii) a patient who availed ATT from the hospital's pharmacy identified through the pharmacy information system (PIS).

Question:Please clarify if the definition of the TB patient was taken as the inclusion criteria for the study.

Authors' Response:*Yes this definition was considered as inclusion criteria for selecting TB cases. We have clarified this on page 8 in paragraph 3.*

What were the exclusion criteria?

Authors' Response:*There were no exclusion criteria.*

Question:Page 8 Line 15: In reference to patient details, please indicate the type of data variables that were extracted for the study participants.

Authors' Response:*Thank you, we have now specified this on page 9 in paragraph 2: Details of patients with TB were extracted from multiple sources and included the patients name, hospital and demographics such as age, residence, education (urban/rural), marital status, year diagnosed, clinical department visited and electronic source of the record.*

Question:Page 9 Line 33: Please indicate how many people were in the focus group discussion (FGD) and the rationale for involving only nurses in the FGD.

Authors' Response: *Since we wanted to capture the notification issues for in-patients we decided to include nursing staff from different wards where patients with TB were admitted and managed. In the study hospital, nursing staff responsible for reporting all disease considered notifiable to the Medical Records Department and TBHV. As the nursing staff, at the hospital are a relatively homogenous group (at the time of the study the nurses comprised only women), we decided to conduct FGD exclusively for the nursing staff. This has been inserted in the qualitative section of the results. **We have stated this on page 10 in paragraph 5.***

RESULTS

Question: In the methods, it is indicated that 11 in-depth interviews were conducted. Yet in the results it is stated that 22 health care staff were interviewed. Please resolve the discrepancy.

Authors' Response: *Thank you for this comment. We have resolved this confusion by clarifying the number of participants from both the in-depth interviews (IDI) and FGD, as follows:*

Paragraph 1 on page 16.

A total of 22 healthcare providers (11 from IDI and 11 from FGD) from various clinical departments at the hospital were interviewed.

Additionally we have clarified the reason for including nursing staff in methods section of the manuscript in on page 10 in paragraph 5.

Question: Page 17 Line 40 Please provide information on the proportion of those interviewed who were aware of the RNTCP requirement of notification.

Authors' Response: *Thank you, we have clarified this.*

Last paragraph on page 19: Out of 22 health care providers 14 were aware of the RNTCP requirement of notification. This has now been included in the results in

Question: Please explain whether for the qualitative results, the statements in italics are statements from the focus groups or in-depth interviews.

Authors' Response: *Thank you! We confirm that the statements in italics are quotes from the participants of the focus groups and in-depth interviews.*

Question: It is not quite clear which is which as some have what appears to be the informant stated which others are just presented. Please organize the results in a way that indicates what emerges from the informant interviews and what is from the FGD.

Authors' Response: *Thanks for this suggestion. The entire qualitative results section comprises informant stated information synthesised into subthemes and overarching themes as per the framework approach put forth by Ritchie J et al[@] (ref #14). Additionally combining FGDs and IDIs form a part of data triangulation in qualitative research and improves the richness of the data. This is also required to achieve a more comprehensive understanding of the issues surrounding notification#. We feel that organization of results or themes according to the method of interview used may not add any insights into the narrative. Hence, we have not made the changes as suggested and sincerely hope that both the reviewer and the editor understand our stance.*

[@] Ritchie J, Lewis J, McNaughton Nicholls C, et al. *Qualitative research practice: a guide for social science students and researchers*. Sage 2014.

[#] Lambert SD and Loiselle CG. *Combining individual interviews and focus groups to enhance data richness*. *J Adv Nur*. 2008. 62(2): 228-37.

DISCUSSION

Question: The discussion highlights important information and recommendations for addressing the gaps in TB notification identified in the study. I am also expecting to find how the study findings

compared with what other studies in India and elsewhere have found. Please address this gap in the discussion.

Authors' Response: *Thank you for this comment! We have tried to compare findings from our study with other studies as best as is possible. Studies on TB notification, especially from the private sector in India are limited. However, if there are still specific gaps existing we are happy to address those.*

Question: Page 22 Lines 5 to 18 I was expecting to see this information in the introduction to put the study within the context of the prevailing situation. I suggest that this section is moved to the Introduction.

Authors' Response: *Thank you, for the comment. We have updated the introduction with this information.*

Page 22 Line 31. Please provide the references

Authors' Response: *As suggested we have included relevant references*

Page 25 Line 44 to Page 26 Line 12 is not clear. What project is being referred? How is this project related to the study. This section is a bit confusing and should be clarified.

Authors' Response: *We have attempted to clarify the difficulty in understanding and have modified the sentences to provide better clarity. This has been given below.*

Page 27, paragraph 3:

Finally, testing for MDR TB cases in the study hospital was done using GenXpert MTB/RIF® equipment that was acquired through the Initiative for Promoting Affordable and Quality TB tests (IPAQT) project. IPAQT aims to bring WHO approved TB tests at affordable prices to patients in private sector. [29,30] This was the only MDR TB diagnostic service available at the hospital. MDR TB cases detected through IPAQT are entered into Nikshay through a sub-portal within the hospital's primary Nikshay portal. It is only through this sub-portal that a person diagnosed with MDR TB at the hospital could be notified. A lack of awareness of this separate portal prevented notification of MDR TB diagnosed in the hospital.

Question: Page 26 Line 31 Sentence is not clear. Please rephrase.

Authors' Response: *We have rephrased this statement as follows in 4 page 28 as follows: As described in the methods section of the manuscript, the participants in the qualitative component were fairly representative of the healthcare providers involved in the management of a patient with TB at the hospital.*

Authors' Response: *Thank you, we have now rephrased this!*

Page 26 Line 43 The informants were said to have been conveniently sampled; therefore to now indicate that those interviewed were fairly representative of the health care staff warrants some more information in the Results section on how those interviewed reflects the population of health staff.

Authors' Response: *Thank you for this comment! We would like to clarify that we used purposive sampling, not a convenient one, which is the norm in qualitative research. The study tried to capture issues pertaining to TB notification and ensured that all groups of healthcare providers and departments that provided care were represented. As far as possible the study included health care providers who encountered patients with TB at possible points in the hospital as represented in Fig 1, including hospital staff and RNTCP staff. Also, we achieved saturation of qualitative findings. Therefore, we believe, based on the qualitative methods that we have used, that this sample is fairly representative of those health care providers who have experience with patients with TB.*

We have described this in the methods section of the manuscript as follows in paragraph 1 page 9: Qualitative component

Health care providers caring for patients with TB from various departments including clinicians, staff nurse, researchers, RNTCP LT, and RNTCP TBHV were interviewed in-depth. Participants were

chosen purposively to include those involved at various points within the TB case management cascade depicted in **Figure 1**.

CONCLUSION

The conclusion is in line with the study findings.

Authors' Response: Thank you very much your valuable comments that have helped a lot in fine tuning the manuscript. We strongly believe that these revisions will help readers better understand the study findings the issues surrounding TB notification in India.

VERSION 2 – REVIEW

REVIEWER	Sally-Ann Ohene Ghana College of Physicians and Surgeons, Fellow, Ghana
REVIEW RETURNED	09-Oct-2018

GENERAL COMMENTS	The authors are commended for addressing the gaps identified in the first review. My suggestion is that the paper undergoes proof-reading to address the few outstanding grammatical errors. Some of the issues have been pointed below and in the paper itself for the authors' consideration. GENERAL REMARKS The authors are commended for addressing the gaps identified in the first review. My suggestion is that the paper undergoes proof-reading to address the few outstanding grammatical errors. Some of the issues have been pointed below and in the paper itself for the authors' consideration. ABSTRACT In the Abstract, there is a reference to "Nikshay" in the Objectives and Conclusion. It is useful to add a phrase to let readers know this is the online TB notification system. BACKGROUND Page 5 Line 26 I suggest the following sentence "However, in 2017 only 19% (81% from public sector) of these patients receive care from, or are notified i.e., reported, to the Revised National Tuberculosis Control Programme (RNTCP)[4,7]" is reworded to "However, in 2017 only 19% of these patients from the private sector were notified to the Revised National Tuberculosis Control Programme (RNTCP)[4,7]" Page 5 Line 26 the phrase "continues to below" should read "continues to be low." METHODS Page 8 Line 34: function as DOT "centers" RESULTS Please maintain one tense in the headings in the Qualitative component section The reviewer also provided a marked copy with additional comments. Please contact the publisher for full details.
---

VERSION 2 – AUTHOR RESPONSE

GENERAL REMARKS

The authors are commended for addressing the gaps identified in the first review. My suggestion is that the paper undergoes proof-reading to address the few outstanding grammatical errors. Some of the issues have been pointed below and in the paper itself for the authors' consideration.

Authors' response- Thanks a lot for those remarks. As per the suggestion, the manuscript was proofread by one of the co-authors, EW whose first language is English. All the grammatical errors have been rectified as far as possible.

ABSTRACT

In the Abstract, there is a reference to "Nikshay" in the Objectives and Conclusion. It is useful to add a phrase to let readers know this is the online TB notification system.

Authors' response- Incorporated

BACKGROUND

Page 5 Line 26 I suggest the following sentence "However, in 2017 only 19% (81% from public sector) of these patients receive care from, or are notified i.e., reported, to the Revised National Tuberculosis

Control Programme (RNTCP)[4,7]" is reworded to "However, in 2017 only 19% of these patients from the private sector were notified to the Revised National Tuberculosis Control Programme (RNTCP)[4,7]"

Page 5 Line 26 the phrase "continues to below" should read "continues to be low.

Authors' response- Corrected

METHODS

Page 8 Line 34: function as DOT "centers"

Authors' response- Corrected

RESULTS

Please maintain one tense in the headings in the Qualitative component section

Authors' response- Corrected

Thank you very much your valuable comments. All revisions have been highlighted in the manuscript.